Towards efficient glaucoma screening with modular convolution-involution cascade architecture

Mouhafid Mohamed 1
Zhou Yatong 1 zyt@hebut.edu.cn
Shan Chunyan 2
Xiao Zhitao 3
1 School of Electronics and Information Engineering, Hebei University of Technology , Tianjin , China
2 NHC Key Laboratory of Hormones and Development, Tianjin Key Laboratory of Metabolic Diseases, Chu Hsien-I Memorial Hospital & Tianjin Institute of Endocrinology, Tianjin Medical University , Tianjin , China
3 School of Life Sciences, Tiangong University , Tianjin , China
Coelho Paulo Jorge
Electronic publication date: 2025 Apr 21
Publication date: 2025
Volume: 11
Electronic Location ID: e2844
Received 2024 Nov 20; Accepted 2025 Mar 27
Copyright: © 2025 Mouhafid et al.
Copyright year: 2025
Copyright holder: Mouhafid et al.
License: This is an open access article distributed under the terms of the Creative Commons Attribution License, which permits unrestricted use, distribution, reproduction and adaptation in any medium and for any purpose provided that it is properly attributed. For attribution, the original author(s), title, publication source (PeerJ Computer Science) and either DOI or URL of the article must be cited.
License URL: https://creativecommons.org/licenses/by/4.0/

Keywords: Computer-aided diagnosis, Precision medicine, Medical imaging, Retinal fundus images

Funding: Beijing Tianjin Hebei Basic Research Cooperation J210008, 21JCZXJC00170, and H2021202008 Tianjin Key Medical Discipline (Specialty) Construction Project TJYXZDXK-032A This work was supported by the Special Foundation for Beijing Tianjin Hebei Basic Research Cooperation (J210008, 21JCZXJC00170, H2021202008) and Tianjin Key Medical Discipline (Specialty) Construction Project (TJYXZDXK-032A). The funders had no role in study design, data collection and analysis, decision to publish, or preparation of the manuscript.

==============================
Automated glaucoma detection from retinal fundus images plays a crucial role in facilitating early intervention and improving the management of this progressive ocular condition. Although convolutional neural networks (CNNs) have significantly advanced image analysis, current CNN-based models encounter two major limitations. First, they rely primarily on convolutional operations, which restrict the ability to capture cross-channel correlations effectively due to the channel-specific focus of these operations. Second, they often depend on fully-connected (FC) layers for classification, which can introduce unnecessary complexity and limit adaptability, potentially impacting overall classification performance. This study introduces the Modular Convolution-Involution Cascade Network (MCICNet), an innovative CNN architecture designed to address these challenges in the context of glaucoma detection. The model employs a combination of convolution and involution operations in a cascade structure, allowing for the effective capture of inter-channel dependencies within the feature extraction process. Furthermore, the classification phase integrates light gradient boosting machine (LightGBM) as a replacement for traditional FC layers, offering enhanced precision and generalization while reducing model complexity. Extensive experiments conducted on the LAG and ACRIMA datasets demonstrate that MCICNet achieves significant improvements compared to existing CNN and transformer-based models. The model attained a classification accuracy of 95.6% on the LAG dataset and 96.2% on ACRIMA, outperforming nine widely used CNN architectures (AlexNet, MobileNetV2, SqueezeNet, ResNet18, GoogLeNet, DenseNet121, EfficientNetB0, ShuffleNet, and VGG16), as well as three transformer-based models (ViT, MaxViT, and SwinT). Additionally, MCICNet showed superior performance over its variant without involution (MCICNet-NoInvolution). With only 0.9 million parameters, MCICNet demonstrates substantial efficiency in resource utilization alongside its high learning capability, establishing it as an advanced and computationally efficient solution for glaucoma detection.

Introduction

Glaucoma is a prevalent ocular disorder recognized for its chronic, irreversible progression. Although therapeutic and surgical strategies can mitigate the advance of the disease, they do not achieve complete cessation, establishing glaucoma as a major contributor to permanent vision loss globally. The World Health Organization notes that more than 65 million individuals worldwide are affected by this condition (Bourne, 2006). The stealthy, asymptomatic nature of glaucoma’s progression necessitates proactive early detection and management to conserve visual function.

The disorder is identified by specific morphological changes in the retinal fundus, focusing on the optic disc (OD) and the retinal nerve fiber layer (RNFL). The OD’s appearance in fundus imagery is crucial for diagnosing glaucoma, with key indicators being the thinning or notching of the optic nerve rim, cupping, an increased cup-to-disc ratio (CDR), disc hemorrhage, and defects within the RNFL. The CDR, which measures the proportion of the optic cup (OC) to the OD area, is a critical diagnostic tool (Chang & Singh, 2016; Gordon et al., 2002). Figure 1 demonstrates that an enlarged OC relative to the OD diameter is suggestive of glaucoma, while a smaller OC is indicative of a healthy eye. These visual signs are essential not only for diagnosing but also for tracking the progression of glaucoma, underlining the necessity for advanced imaging techniques to detect nuanced structural changes early and accurately.

Figure 1 Comparison of retinal fundus images between glaucomatous and non-glaucomatous eyes, highlighting the differences in OC size as a diagnostic feature of glaucoma.

The emergence of deep learning (DL), particularly convolutional neural networks (CNNs), has significantly advanced the field of medical image analysis, enabling the automation of glaucoma detection from fundus images (Geetha et al., 2025; Saqib et al., 2024; Velpula et al., 2024). Despite their potential, current CNN-based approaches are constrained by two major limitations that impact their diagnostic accuracy and practical applicability. The first limitation arises from the reliance of conventional CNNs on convolutional operations that are primarily designed for spatial feature extraction (Diaz-Pinto et al., 2019; Esengönül & Cunha, 2023; Raj et al., 2024). Although these operations are effective at capturing localized patterns, they struggle to model cross-channel correlations due to the use of channel-specific kernel weights. This constraint is particularly critical in glaucoma diagnosis, where important biomarkers (e.g., CDR progression, RNFL defects) often arise from subtle interactions between color channels and anatomical structures. For example, the pallor of the OC in red-free imaging or the contrast between neuroretinal rim tissues across different spectral channels may contain essential diagnostic information that standard convolution operations fail to capture in an integrated manner.

The second limitation involves the use of FC layers in many CNN architectures for final classification (Devecioglu et al., 2021; Serte & Serener, 2019). FC layers, while common, introduce several drawbacks, such as the loss of spatial relationships within feature maps, the compression of high-dimensional representations into fixed weight matrices, and the substantial increase in model parameters. These factors collectively contribute to an elevated risk of overfitting and reduce the model’s ability to adapt to intricate pathological features. This challenge is particularly pronounced in medical imaging applications like glaucoma detection, where limited training data heightens the susceptibility of FC-based models to performance deterioration.

These limitations give rise to two critical research questions: (1) How can DL models be redesigned to effectively capture both spatial features and cross-channel dependencies that are vital for identifying glaucoma biomarkers? (2) Can alternative classification strategies replace FC layers to improve model efficiency and diagnostic accuracy in contexts with limited data availability? Addressing these questions is crucial for bridging the gap between theoretical advancements in model development and the practical needs of clinical diagnostics.

This study introduces the MCICNet, a novel architecture that addresses these challenges. The main contributions of this article can be summarized as follows: Proposes a novel CNN architecture, MCICNet, for automated glaucoma detection from retinal fundus images.

Introduces a modular convolution-involution cascade structure to enhance cross-channel feature extraction, addressing limitations of traditional convolutional layers.

Replaces conventional FC layers with LightGBM in the classification phase, improving precision and model efficiency.

Demonstrates the superior performance of MCICNet compared to nine benchmark CNN models, three transformer-based architectures, and a variant without involution, achieving classification accuracies of 95.6% on the LAG dataset and 96.2% on ACRIMA.

Reduces computational complexity with a lightweight model of only 0.9 million parameters, offering a resource-effective solution for glaucoma detection.

Related works

Glaucoma detection from retinal fundus images has gained substantial interest, with numerous studies investigating both traditional machine learning (ML) and DL methods. These studies can be broadly classified into two categories: classification-only approaches and those integrating segmentation with classification.

Earlier methods focused on handcrafted features and conventional classifiers. Dua et al. (2011) explored wavelet features using Daubechies, Symlets, and Biorthogonal filters, selecting the most effective subsets via 2D Discrete Wavelet Transform (DWT) for energy signatures. They used classifiers like support vector machine (SVM), random forest (RF), and naïve Bayes (NB), reaching 93% accuracy with tenfold cross-validation. Singh et al. (2016) advanced this by combining wavelet features with genetic optimization on segmented OD without blood vessels, achieving 94.7% accuracy. Acharya et al. (2011) combined texture and Higher-Order Spectra (HOS) features, obtaining 91% accuracy, emphasizing clinical relevance. Krishnan & Faust (2013) developed a cost-effective system with hybrid features from fundus images, including HOS, trace transform, and DWT, and reported high accuracy, sensitivity, and specificity using SVM with different kernels.

With the advent of DL, researchers shifted toward CNNs. Elangovan & Nath (2021) developed a DL architecture using a custom 18-layer CNN for glaucoma diagnosis from fundus images. This CNN features four convolutional, two max-pooling, and one FC layer. Performance optimization involved multi-stage parameter tuning and data augmentation through rotation, achieving 94.43% accuracy on the LAG dataset. Chen et al. (2015) created a DL model with a six-layer CNN, including dropout and data augmentation, for glaucoma diagnosis. Experiments on ORIGA and SCES datasets resulted in AUC values of 0.831 and 0.887. Li et al. (2019) introduced an attention-based CNN for glaucoma, focusing on local pathology areas in fundus images to improve both accuracy and interpretability, using the LAG dataset. Gómez-Valverde et al. (2019) explored DL factors like dataset size and architecture, comparing transfer learning (TL) with VGG19 against human experts. Their approach achieved an AUC of 0.94 across datasets, showing promise for CAD systems in glaucoma screening. Christopher et al. (2018) analyzed a diverse database of retinal scans, where DL models distinguished glaucoma with an AUC of 0.91 overall, and specifically 0.97 for moderate-to-severe and 0.89 for mild functional loss cases. Pal, Moorthy & Shahina (2019) proposed G-EyeNet, a hybrid DL architecture combining a convolutional autoencoder and a CNN classifier for early glaucoma detection from fundus images, optimizing for both reconstruction and classification errors. Al-Bander et al. (2017) used a hybrid system of CNN for feature learning and SVM for classification, achieving 88.2% accuracy, 90.8% specificity, and 85% sensitivity in glaucoma classification from retinal images. Fan et al. (2023) conducted a comparison between the DeiT transformer and ResNet50 using six datasets. Both models achieved similar AUC scores (0.82–0.91) on the OHTS data, but the DeiT transformer showed superior generalization across external datasets.

In contrast to classification-only methods, some studies have integrated segmentation into their glaucoma detection models. Nayak et al. (2009) used pre-processing, morphological operations, and thresholding to detect OD and blood vessels automatically. They extracted features like CDR, distance ratios, and vessel area ratios, validating them with neural network classification, showing effective feature extraction via thresholding. Issac, Partha Sarathi & Dutta (2015) concentrated on glaucoma-specific parameters like CDR and neuro-retinal rim area, segmenting OD and OC using adaptive pixel intensity thresholding. Their method was robust against image quality issues, achieving 94.11% accuracy, underscoring the value of thresholding in glaucoma diagnosis. Bajwa et al. (2019) proposed a two-step approach for OD detection and glaucoma diagnosis with regions-with-CNN (RCNN) for OD identification and a custom CNN for classification. They developed a semi-automatic rule-based technique for ground truth data due to the lack of annotations, resulting in an AUC of 0.874 on the ORIGA dataset. Gao et al. (2024) created an automated system using YOLOv7 for OD and OC localization, achieving a Pearson correlation of 0.91 and a mean absolute error of 0.0347, proving the effectiveness of YOLO for glaucoma feature detection. Veena, Muruganandham & Senthil Kumaran (2022) introduced a dual-CNN for segmenting OD and OC, calculating CDR for glaucoma detection, with 98% accuracy for OD and 97% for OC segmentation on the DRISHTI-GS dataset, highlighting deep learning’s utility in retinal segmentation. Li et al. (2022) developed an end-to-end RCNN for joint OD and OC segmentation, incorporating attention modules, which performed well across multiple datasets. Hervella et al. (2022) employed a multi-task learning approach for both segmentation and classification, sharing learned representations to achieve robust performance without complex adjustments. Neto et al. (2022) enhanced U-Net with TL from Inception models for segmentation, achieving high Dice coefficients and AUC for glaucoma screening. Nawaz et al. (2022) used EfficientDet-D0 with EfficientNet-B0 for OD and OC lesion detection, showcasing superior accuracy on the ORIGA dataset and good generalization across datasets. Desiani et al. (2023) applied a two-stage method with U-Net for OD segmentation followed by Xcep-Dense for classification, achieving over 87% accuracy on the LAG dataset.

Despite advances in automated glaucoma detection, current methods face challenges that limit their effectiveness and scalability. Traditional CNNs, reliant on convolutional operations (Elangovan & Nath, 2021; Chen et al., 2015; Li et al., 2019; Gómez-Valverde et al., 2019; Christopher et al., 2018; Pal, Moorthy & Shahina, 2019; Al-Bander et al., 2017; Nayak et al., 2009; Bajwa et al., 2019; Hervella et al., 2022; Nawaz et al., 2022), struggle to capture inter-channel correlations due to their channel-specific nature, compromising feature extraction in complex medical imaging where such dependencies are vital. Additionally, FC layers in classification (Elangovan & Nath, 2021; Chen et al., 2015; Li et al., 2019; Gómez-Valverde et al., 2019; Christopher et al., 2018; Pal, Moorthy & Shahina, 2019; Al-Bander et al., 2017; Bajwa et al., 2019; Hervella et al., 2022) add complexity, heightening overfitting risks and reducing generalization, which impairs performance in large-scale screening across diverse datasets. Furthermore, the large parameter counts in DL models (Gómez-Valverde et al., 2019; Christopher et al., 2018; Al-Bander et al., 2017; Neto et al., 2022) inflate complexity and training time, demanding significant computational resources impractical for real-time, resource-limited settings, while also obscuring interpretability. These issues underscore the need for an efficient approach that captures inter-channel dependencies without sacrificing computational feasibility. MCICNet addresses this with a novel architecture blending convolution and involution in a cascading design, enhancing feature extraction and detection of subtle glaucomatous changes in retinal images. Replacing FC layers with LightGBM further reduces complexity and boosts generalization.

Materials and methods

This article introduces MCICNet, a hybrid architecture with innovative modules for feature extraction. It starts with the residual multi-scale feature fusion module (RMSFFM) for multi-scale spatial features, followed by cascading residual hybrid convolutional-involutional module (RHCIM) layers using depth-wise, dilated convolutions, and involution operations with residual connections and tuned parameters for efficient medical image processing. LightGBM replaces CNN FC layers for better classification accuracy. Figure 2 shows MCICNet’s structure.

Figure 2 Structure of the proposed MCICNet.

Residual multi-scale feature fusion module

The RMSFFM enhances feature extraction for glaucoma detection by integrating multi-scale information, accommodating varied pathological signs. Key components include: Multi-size convolutions: Applies 1 × 1, 3 × 3, and 5 × 5 kernels to input feature maps. 1 × 1 adjusts channel relationships, 3 × 3 balances local spatial data, and 5 × 5 captures broader spatial context.

Feature concatenation: Combines convolution outputs along channels, merging multi-scale spatial details into a unified feature map for subsequent processing.

Feature fusion: Uses a 1 × 1 convolution post-concatenation to integrate multi-scale data, reduce dimensionality, and prepare features for later network stages.

Residual connection: Merges the original input with the fused output via a skip connection, preserving input data integrity, mitigating vanishing gradients, and supporting deep network training.

Residual hybrid convolutional-involutional module

The RHCIM is a composite framework that combines depth-wise and dilated convolutions with involution operations, supported by residual connections. This structure leverages the efficiency of depth-wise convolutions, the broadened receptive field of dilated convolutions, and the adaptive capabilities of involution. The integration of these hybrid convolutional and involutional techniques within RHCIM represents a key contribution of this work.

Dilated convolution sub-module

Figure 3 illustrates a typical convolution process. Imagine that an input’s feature maps have heights and widths represented by H and W. These maps are symbolized as X in a space defined by the product of height, width, and the number of input channels Ci. We refer to a set of Co convolution filters by F∈RCo×Ci×K×K, where each filter is a collection of Ci kernels, each of size K by K. Each kernel within a filter is represented by Fk∈RCi×K×K, where k ranges from 1 to Co, and each kernel element is denoted by Fk,c∈RK×K, with c ranging from 1 to Ci. To derive the output feature maps, designated as Y∈RH×W×Co, these convolution filters are systematically applied to the input maps. This process involves sliding the filters over the input and conducting element-wise multiplications followed by a summation, as defined in Eq. (1).

(1) Yi,j,k=∑c=1Ci⁡∑(u,v)∈δ⁡Fk,c,u+⌊K/2⌋,v+⌊K/2⌋Xi+u,j+v,c.

Figure 3 Representation of the convolution module.

In this context, δ∈Z2represents the set of offsets in the neighborhood considered during convolution around the position (u,v), expressed as:

(2) δ=[−⌊K/2⌋,…,⌊K/2⌋]×[−⌊K/2⌋,…,⌊K/2⌋].

Within this convolution module, the dilation operation is executed by inserting zeros between filter elements. This dilation technique enables the network to encompass more pertinent information by expanding the receptive field of the filters. Consider a filter F of size K×K applied to an input with a dilation rate d. Instead of covering a K×K area of the input, the filter now spans an area of [K+(K−1)(d−1)]×[K+(K−1)(d−1)], with (d−1) zeros inserted between each original element in the filter. This does not change the number of parameters, as the inserted values are zeros, but it does allow the filter to aggregate information over a broader area of the input. Our experiment utilized a dilated convolution within the RHCIM, employing a filter size of 3 × 3.

Involution sub-module

There’s a known issue with convolution operations in neural networks, where channels can be overly dependent on each other, causing a lack of adaptability. To improve this, a new type of kernel, called an involution kernel H∈RH×W×K×K×G, has been designed to change the way transformations are handled in both the spatial and channel domains (Fig. 4). For each pixel Xi,j∈RC located at ( i, j), the involution kernel Hi,j,.,.,g∈RK×K(g=1,2,…,G) is customized, although it remains consistent across channels. A feature map is then created from the input by applying these involution kernels through a series of multiplications and additions. The variable G represents the number of groups, with each group utilizing the same involution kernel. A feature map (expressed in Eq. (3)) is then created from the input by applying these involution kernels through a series of multiplications and additions.

(3) Yi,j,k=∑(u,v)∈δ⁡Hi,j,u+⌊K/2⌋,v+⌊k/2⌋,⌈kG/C⌉Xi+u,j+v,k.

Figure 4 Representation of the involution module.

In contrast to convolution kernels, the configuration of involution kernels H is contingent on the input feature map X. Alignment of the output kernels with the input is achieved by generating involution kernels based on the original input tensor. Consequently, the kernel generation function ϕ and the mapping function at every position ( i, j) can be formulated as follows:

(4) Hi,j=ϕ(Xψi,j).

Here, ψi,j represents the set of pixels that Hi,j depends on. The kernel generation function ϕ:RC→RK×K×G with ψi,j={(i,j)} is outlined as presented in Eq. (5).

(5) Hi,j=ϕ(Xi,j)=W1σ(W0Xi,j).

In this context, W0 belonging to RCr×Cand W1 belonging to R(K×K×G)Cr represent two linear transformations that jointly form a bottleneck structure. The reduction in channels, governed by a ratio r, is utilized to enhance computational efficiency. Additionally, the symbol σ signifies the inclusion of batch normalization and non-linear activation functions between the two linear projections. The process of feature generation described in Eq. (3) can be seen as an extended version of self-attention (Vaswani et al., 2017). In self-attention, values V are pooled according to affinities determined by calculating the similarity between the query Q and the key K, as expressed in Eq. (6).

(6) Yi,j,k=∑(p,q)∈δ⁡(QKT)i,j,p,q,⌈kH/C⌉Vp,q,k.

In this scenario, Q, K, and V experience linear transformations from the input X, while H represents the number of heads in multi-head self-attention (Vaswani et al., 2017). The similarity arises from both operators gathering pixels within the neighborhood ϑ or a more loosely bounded range via a weighted sum. Regarding involution computation, it can be interpreted as a form of spatial attentive aggregation. Conversely, the attention map (also termed affinity or similarity matrix) QKT in the self-attention mechanism can be seen as a variant of the involution kernel H.

In Fig. 4, the involution kernel Hi,j∈RK×K×1 (where G=1 for simplicity in demonstration) is generated from the function ϕ utilizing a single pixel at position ( i, j), after which a rearrangement from channel to space is performed. The involution’s multiply-add operation encompasses two stages: N, indicating multiplication broadcasted across C channels, and L, signifying summation aggregated within the K×K spatial neighborhood.

Depth-wise convolutional sub-module

The depth-wise convolutional layer in MCICNet enhances computational efficiency and simplifies the model while retaining the ability to capture key spatial features. It uses depth-wise separable convolution, consisting of depth-wise and pointwise convolution operations.

In the initial step, the module executes depth-wise convolution, wherein a single convolutional filter is individually applied to each input channel. Unlike standard convolution, which merges all input channels and filters them collectively to generate multiple output channels, depth-wise convolution keeps these operations distinct. This implies that if the input data has Ci channels and the kernel size is K×K, the depth-wise convolution will employ Ci different K×K filters on each of the Ci channels. The primary advantage of this approach lies in its efficiency. Because each filter operates on a single channel, the number of multiplications decreases significantly compared to traditional convolution operations. Specifically, for an input of size H×W×Ci and a filter of size K×K, the depth-wise convolution requires H×W×Ci×K2 multiplications. This is Ci times less than the standard convolution, which would require H×W×Ci×K2×C0 multiplications for C0 output channels.

After the depth-wise convolution, the module utilizes a pointwise convolution, which essentially involves a 1 × 1 convolution. In this step, the output of the depth-wise convolution is combined across channels. The pointwise convolution operates on each pixel individually, taking the Ci channels from the depth-wise step and merging them into C0 output channels. This operation is responsible for generating a new set of features by linearly combining the outputs of the depth-wise convolution.

Cascade deployment of RHCIM

MCICNet applies the RHCIM in a sequentially layered structure, with each configuration progressively optimized to advance feature extraction processes: Initial RHCIM layer: Utilizes 32 filters to amplify multi-scale features from the RMSFFM, initiating sophisticated spatial and channel-based processing as the network’s foundational layer.

Subsequent RHCIM layers: Deeper layers increase filter counts (64, 128, 256), refining feature maps and enabling extraction of complex features as network depth grows.

Involution parameter adjustments: Channel sizes adjust (16, 64, 128, 256) to match filter counts, maintaining adaptive involution kernels responsive to evolving feature maps.

Depth multiplier and dilation rate: Advanced layers raise the depth multiplier to 2 and dilation rate to 4, expanding the receptive field to capture intricate spatial patterns effectively.

Advanced feature analysis with LightGBM

In the proposed MCICNet framework, the extraction of high-level feature maps is a crucial step for the subsequent classification task. After passing through the various layers of the MCICNet architecture, the network generates feature maps that encode essential spatial and contextual information from the input image. These feature maps are then flattened into a one-dimensional vector for each input sample, preparing them for the classification stage. Unlike conventional CNNs that rely on FC layers for classification, the proposed MCICNet employs LightGBM as an advanced classifier to handle the extracted feature vectors. This choice is motivated by LightGBM’s ability to efficiently manage high-dimensional data while maintaining accuracy (Ke et al., 2017).

LightGBM builds decision trees sequentially, with each tree correcting errors from the previous ones. It initializes predictions to zero and iteratively refines them through boosting rounds. In each round, residuals (differences between true labels and current predictions) are calculated, and a new decision tree is trained to predict these residuals. Predictions are updated by adding a fraction of the tree’s output, scaled by a learning rate. After multiple rounds, final predictions are determined using a threshold function. This approach allows MCICNet to handle complex classification tasks efficiently, avoiding the computational overhead of FC layers. The hybrid MCICNet-LightGBM model is outlined in Algorithm 1, available in the Supplemental Files.

Comparative design framework

To provide a comprehensive evaluation of the MCICNet model’s performance, this section details the methodology employed to systematically compare it with established models.

For comparative analysis, nine prominent CNN architectures were chosen as benchmarks against MCICNet: SqueezeNet, AlexNet, MobileNetV2, DenseNet121, ResNet18, GoogLeNet, ShuffleNet, EfficientNetB0, and VGG16. Additionally, three transformer-based architectures were evaluated for their performance in this glaucoma detection task: Vision Transformer (ViT), MaxViT, and Swin Transformer (SwinT).

For consistent evaluation, all models were tested with an input size of 224 × 224, except for SqueezeNet, which required an input size of 227 × 227. This input size is commonly used in image classification tasks and ensures compatibility across the models. All models were trained using the Adam optimizer with a learning rate of 0.001. No hyperparameter optimization was conducted in this study, and the models were trained with their default architectural settings, aside from the specified optimizer and learning rate, to maintain consistency in the comparison.

To test the hybrid Involution-Convolution method in MCICNet, a structurally identical model replaced involution layers with convolution layers in the RHCIM, creating MCICNet-NoInvolution. Filter numbers and kernel sizes aligned with the dilated convolution setup. This examined the benefits of involution’s adaptive, spatially sensitive kernels versus convolution’s fixed kernels in capturing complex spatial data patterns.

Dataset description and preprocessing techniques

This study used the Large-scale Attention-based Glaucoma (LAG) dataset, containing 5,824 retinal fundus images (2,392 glaucoma, 3,432 healthy) from Beijing Tongren Hospital. Standardized at 500 × 500 pixels, the JPEG images were evaluated by glaucoma experts using morphological (OD, RNFL) and functional (IOP, visual field) assessments, with manual OD reviews for diagnostic accuracy. Each image was labeled as glaucoma or non-glaucoma. A subset of 4,854 images is available upon request, and the dataset can be downloaded at https://github.com/smilell/AG-CNN.

To further assess the effectiveness of the proposed model, an additional dataset, ACRIMA, was utilized (Diaz-Pinto et al., 2019). This dataset consists of 705 retinal fundus images, comprising 396 glaucomatous cases and 309 normal healthy images. These images were acquired from both the left and right eyes and were previously dilated and centered on the OD. The retinal images have a resolution of 2048 × 1536 pixels. The dataset was examined and annotated by two ophthalmologists at the Foundation Ophthalmological Mediterranean. The ACRIMA dataset is available for download at https://figshare.com/s/c2d31f850af14c5b5232.

Both datasets were partitioned into three subsets to ensure an effective training and evaluation process. Specifically, 70% of the data was allocated to the training set for optimizing the model’s parameters, while 15% was designated as the validation set to monitor performance and guide model adjustments during training. The remaining 15% was reserved as the test set to evaluate the model’s generalization on previously unseen data. It is important to note that the 70:15:15 split is at the image level and is a standard practice commonly used in ML. This split ratio is chosen to ensure a balanced distribution of data for training, validation, and testing, which is crucial for training robust models and evaluating their performance accurately. The 70% training set provides a sufficiently large dataset to train the model effectively, while the 15% validation and test sets offer adequate samples to monitor and evaluate the model’s performance, ensuring that the model generalizes well to unseen data.

Moreover, to enhance model training stability and consistency, pixel normalization was performed on each image, scaling pixel values from the original range of 0 to 255 to a range of 0 to 1 by dividing each pixel by 255.0. This normalization step aids the model in processing image data more efficiently by mitigating variations in brightness and contrast, thus promoting optimal network performance and improving generalization across the dataset. Additionally, all images were resized to 96 × 96 pixels to reduce computational demands, while maintaining sufficient resolution for model training. Both MCICNet and MCICNet-NoInvolution were tested with this smaller input size, establishing a model structure that is computationally efficient without compromising performance. Lastly, image enhancement and data augmentation techniques were not applied to preserve the original characteristics of the images, ensuring a more realistic assessment of the model’s performance on raw data.

Evaluation metrics

The proposed model’s performance is assessed quantitatively using metrics such as accuracy, sensitivity, precision, specificity, and F1-score, as outlined in Eqs. (7)–(11):

(7) accuracy=TP+TNFP+FN+TP+TN

(8) sensitivity=TPTP+FN

(9) precision=TPTP+FP

(10) specificity=TNTN+FP

(11) F1=2×PREC×SENPREC+SEN.

Here, true positives (TP) correctly identifies glaucoma cases, false positives (FP) misclassifies healthy individuals as glaucomatous, false negatives (FN) misses glaucoma cases, and true negatives (TN) accurately confirms non-glaucoma cases. Additionally, the Matthews correlation coefficient (MCC) is included as a more balanced metric, particularly useful when dealing with imbalanced datasets. The MCC is defined as:

(12) MCC=TP×TN−FP×FN(TP+FP)(TP+FN)(TN+FP)(TN+FN).

Experimental environment and parameter setting

The training setup for glaucoma detection was established on a dual NVIDIA Tesla T4 GPU configuration, each offering 15 GB of memory, supplemented by 29 GB of system RAM to maintain stable and efficient training processes. All models were implemented and trained using Python version 3.10. TensorFlow with Keras was employed for constructing the feature extraction component of MCICNet, while the LightGBM algorithm was implemented using the Scikit-learn library.

LightGBM parameters were tuned using GBDT for pattern capture. A 0.1 learning rate, 200 boosting rounds, and 31 leaves per tree balanced performance and generalization. Tree depth was unrestricted, with 20 samples minimum per leaf to curb overfitting. Feature and bagging fractions were 1.0 for full data use, and class weights auto-adjusted for imbalance.

For the additional benchmark models, pre-trained weights from ImageNet were utilized to support TL. Input images were resized according to each model’s specific input dimensions and normalized based on ImageNet’s mean and standard deviation. The Adam optimizer was applied for weight updates during training, with a batch size of 32 over a training span of 200 iterations. Additionally, a fixed random seed (set to 7) was used to ensure the reproducibility of the training results.

Results

Accurate glaucoma detection is critical in ophthalmology, requiring a precise and comprehensive analysis of retinal fundus images. This section presents the effectiveness of the proposed MCICNet architecture in addressing this complex task. A detailed evaluation was performed to compare the model’s performance against MCICNet without involution (MCICNet-NoInvolution), as well as and nine convolution-based CNN models and three transformer-based models.

Convergence analysis

Figure 5 illustrates the progression of training and validation accuracy throughout the training process for MCICNet, MCICNet-NoInvolution, AlexNet, MobileNetV2, SqueezeNet, ResNet18, GoogLeNet, DenseNet121, EfficientNetB0, ShuffleNet, VGG16, ViT, MaxViT, and SwinT. A significant observation at the 200th iteration is the superior performance of MCICNet, achieving a final accuracy of 96.2%, followed closely by MCICNet-NoInvolution at 95.6%. This highlights the substantial improvement in learning efficacy achieved through the incorporation of involution operations within the network. In contrast, all other models, including high-performing architectures such as VGG16, recorded final accuracies between 86.4% and 95%. Among the transformer-based models, ViT, MaxViT, and SwinT demonstrated relatively moderate performance, with final validation accuracies below 93%. ViT outperformed MaxViT and SwinT, likely due to its more effective handling of spatial dependencies, while MaxViT’s reliance on feature pooling and SwinT’s hierarchical approach may not have aligned as well with the dataset characteristics. ViT also surpassed most CNN-based models, such as SqueezeNet, ResNet18, and GoogLeNet, except for VGG16 and the two variants of MCICNet. This performance difference can be attributed to ViT’s attention mechanism, which allows the model to focus on relevant regions of an image, thereby improving performance, particularly when global context is important.

Figure 5 Training and validation accuracy curves for various DL models and MCICNet variants.

With regard to stability and convergence, MCICNet demonstrates smoother validation accuracy curves with minimal fluctuations compared to models such as MobileNetV2 and GoogLeNet, which exhibit greater variability across epochs. This stability suggests that MCICNet is resilient to training fluctuations, enhancing its likelihood of rapid and reliable convergence. Furthermore, MCICNet’s fast convergence over epochs suggests an efficient learning process, potentially due to its multi-scale feature fusion and hybrid convolution-involution architecture, which effectively captures pertinent features in fundus images. Conversely, models such as SqueezeNet and ShuffleNet demonstrate a slower learning progression, indicating that they may require additional epochs to achieve optimal performance or may not be as well-suited to the glaucoma detection task as MCICNet.

Several models, including AlexNet, MobileNet, ResNet, GoogLeNet, DenseNet, EfficientNet, ShuffleNet, VGG16, ViT, MaxViT, and SwinT, exhibit a noticeable disparity between training and validation accuracies, indicating overfitting. In these cases, the models perform substantially better on the training data than on the validation data. In contrast, models such as MCICNet-NoInvolution and MCICNet-Involution demonstrate minimal overfitting, as their training and validation accuracies are closely aligned, suggesting improved generalization to unseen data. Certain models, particularly those more prone to overfitting, display fluctuations in validation accuracy, which indicates heightened sensitivity to the specific characteristics of the training data and a reduced ability to generalize to new data.

Performance evaluation

To illustrate the effectiveness of the proposed MCICNet, the metrics precision, sensitivity, specificity, F1-score, accuracy, MCC, and AUC were calculated for both the validation and test datasets, as presented in Tables 1 and 2, respectively. The proposed MCICNet demonstrates superior performance, achieving the highest accuracy (96.29% on the validation set and 95.47% on the test set) and AUC (0.9933 on the validation set and 0.9914 on the test set) among all models, including both older and newer architectures. This is especially significant given that MCICNet is a novel architecture designed specifically for glaucoma detection, utilizing a combination of convolution and involution operations. In comparison, MCICNet-NoInvolution, which replaces involution layers with traditional convolution layers, also performs well but slightly underperforms relative to MCICNet. This indicates that the inclusion of involution operations enhances the model’s ability to capture cross-channel dependencies and spatial features, thus improving performance. Among traditional CNN models, VGG16 ranks as the second-best performer, despite being the oldest model in the comparison after AlexNet. VGG16 achieves 95.06% accuracy on the validation set and 93.55% on the test set, outperforming more recent architectures such as ResNet, DenseNet, SqueezeNet, and EfficientNet. This result is noteworthy, as it is somewhat unexpected for older models like VGG16 to surpass newer architectures. The high accuracy of VGG16 suggests that certain architectural elements may still be relevant in glaucoma detection, even when compared to more recent innovations in CNN design. Transformer-based models, including ViT, MaxViT, and SwinT, demonstrate moderate performance, with SwinT achieving the highest test accuracy (93.55%) among the three. Despite their success in natural language processing and certain computer vision tasks, transformers’ performance in glaucoma detection does not match that of MCICNet. One possible explanation for this is that transformer models typically require large-scale datasets to realize their full potential. While the LAG dataset is substantial, it may not be sufficient to exploit the complete capabilities of transformers, limiting their performance for this specific task. The generalization ability of a model, assessed through its performance consistency across validation and test sets, is crucial for real-world applicability. MCICNet exhibits strong generalization, with only a slight performance drop between the validation and test datasets. Specifically, the accuracy decreases from 96.29% to 95.47%, and the AUC drops from 0.9933 to 0.9914. These minimal reductions suggest that MCICNet is robust and can maintain high accuracy on unseen data, a valuable attribute for clinical applications where reliable and consistent results are essential.

Table 1 Comparison of precision, sensitivity, specificity, F1-score, accuracy, MCC, and AUC metrics on validation set of LAG dataset.

Model	Precision	Sensitivity	Specificity	F1-score	Accuracy	MCC	AUC	
SqueezeNet	0.8539	0.8845	0.8157	0.8590	0.8641	0.7375	0.9590	
AlexNet	0.8629	0.8818	0.8686	0.8700	0.8779	0.7443	0.9547	
MobileNetV2	0.8941	0.8920	0.9280	0.8930	0.9026	0.786	0.9577	
DenseNet121	0.8901	0.9108	0.8877	0.8978	0.9039	0.8031	0.9682	
ResNet18	0.8956	0.9023	0.9174	0.8987	0.9067	0.8006	0.9628	
GoogLeNet	0.9028	0.9056	0.9280	0.9042	0.9122	0.8062	0.9669	
ShuffleNet	0.9092	0.9070	0.9386	0.9081	0.9163	0.816	0.9772	
EfficientNetB0	0.9167	0.9270	0.9280	0.9214	0.9272	0.8457	0.9718	
VGG16	0.9459	0.9459	0.9620	0.9459	0.9506	0.8920	0.9879	
ViT	0.9202	0.9245	0.9386	0.9223	0.9286	0.8240	0.9756	
MaxViT	0.8886	0.8851	0.9260	0.8868	0.8971	0.7700	0.9652	
SwinT	0.9147	0.9139	0.9410	0.9143	0.9218	0.8230	0.9808	
MCICNet-NoInvolution	0.9527	0.9510	0.9682	0.9518	0.9561	0.9040	0.9907	
Proposed MCICNet	0.9591	0.9599	0.9703	0.9595	0.9629	0.9192	0.9933	

Table 2 Comparison of precision, sensitivity, specificity, F1-score, accuracy, MCC, and AUC metrics on test set of LAG dataset.

Model	Precision	Sensitivity	Specificity	F1-score	Accuracy	MCC	AUC	
SqueezeNet	0.8622	0.8935	0.8220	0.8675	0.8724	0.7549	0.9731	
AlexNet	0.8644	0.8837	0.8686	0.8715	0.8792	0.7477	0.9467	
MobileNetV2	0.8953	0.8902	0.9322	0.8927	0.9026	0.7872	0.9712	
DenseNet121	0.9057	0.9187	0.9153	0.9114	0.9176	0.8244	0.9684	
ResNet18	0.8966	0.8959	0.9280	0.8962	0.9053	0.7923	0.9683	
GoogLeNet	0.8898	0.8833	0.9300	0.8864	0.8971	0.7680	0.9578	
ShuffleNet	0.9063	0.9077	0.9322	0.9070	0.9149	0.8140	0.9699	
EfficientNetB0	0.9104	0.9090	0.9386	0.9097	0.9176	0.8100	0.9671	
VGG16	0.9291	0.9298	0.9492	0.9294	0.9355	0.8592	0.9793	
ViT	0.9262	0.9231	0.8949	0.9246	0.9314	0.8493	0.9790	
MaxViT	0.9169	0.9075	0.8638	0.9119	0.9204	0.8244	0.9707	
SwinT	0.9297	0.9289	0.9066	0.9293	0.9355	0.8587	0.9808	
MCICNet-NoInvolution	0.9493	0.9452	0.9682	0.9472	0.9519	0.8934	0.9887	
Proposed MCICNet	0.9515	0.9491	0.9682	0.9503	0.9547	0.9000	0.9914	

ROC and AUC analysis

ROC curves were employed to assess the efficacy of the proposed MCICNet on both the validation (Fig. 6A) and test sets (Fig. 6B). The ROC curve graphically represents the relationship between TPR and FPR across various threshold values. On the validation set, MCICNet achieved a high AUC of 0.9933, with its ROC curve closely following the top-left boundary of the ROC space, signifying a robust TPR with minimal FPR at multiple threshold points. Similarly, MCICNet-NoInvolution also demonstrated strong performance with an AUC of 0.9907, only marginally lower than MCICNet with involution. Although the VGG16 architecture, with its substantial parameter count, achieved a competitive AUC of 0.9879, it was outperformed by the MCICNet models. Noteworthy models also include EfficientNetB0 and ShuffleNet, both of which yielded AUC values above 0.97, whereas AlexNet exhibited lower efficacy (AUC = 0.9547). On the test set, the proposed MCICNet continued to demonstrate leading performance, achieving an AUC of 0.9914, underscoring its generalizability to previously unseen glaucoma images. Similarly, MCICNet-NoInvolution maintained a high AUC of 0.9887, highlighting the effectiveness of the modular cascading design. While the remaining models displayed stable performance, they did not match the MCICNet architectures in accuracy.

Figure 6 ROC curves for various DL models and MCICNet variants.

(A) Validation set; (B) Test set.

External dataset evaluation with seed averaging for stability

To validate the robustness and generalizability of the MCICNet model, an external evaluation was conducted using the ACRIMA dataset. This step was critical to confirm that MCICNet’s performance was not solely dependent on the characteristics of the LAG dataset used for training and initial validation. Figure 7 illustrates the accuracy and loss curves for the training and validation phases on the ACRIMA dataset. Here, MCICNet showed commendable performance; the training and validation accuracy curves in Fig. 7A quickly converged to high levels, with validation accuracy closely tracking the training curve, indicating no significant overfitting. The training accuracy neared 100%, while the validation accuracy stabilized at 97.14%, demonstrating good generalization to new data. The loss curves in Fig. 7B similarly declined sharply and then stabilized, with minimal divergence between training and validation losses, further attesting to the model’s robustness.

Figure 7 Learning curves for training and validation on ACRIMA dataset.

(A) Accuracy; (B) Loss.

The stability and generalizability were further explored through seed averaging experiments on the LAG and ACRIMA datasets, with findings reported in Tables S1 and S2. These experiments used two different seeds (Seed A and Seed B) for model initialization, aiming to ensure performance consistency despite random weight initialization. For the LAG dataset, MCICNet achieved an average accuracy of 95.61% with negligible variance between seeds. On the ACRIMA dataset, accuracy was consistent across seeds as well. This stability across different metrics like precision, sensitivity, specificity, and AUC underscores MCICNet’s reliability for medical imaging applications.

A notable aspect of this study is the model’s capability to generalize across datasets with different characteristics. The ACRIMA dataset, unlike LAG, features pre-cropped OD regions instead of full fundus images. Despite this, MCICNet delivered an average accuracy of 96.26%, precision of 99.34%, and sensitivity of 96.09%, indicating strong adaptability.

MCICNet consistently outshone other models across both datasets. On the LAG dataset, it achieved a superior specificity of 95.11% compared to VGG16 and transformer models like ViT. On ACRIMA, it achieved perfect specificity at 97.50%, contrasting with lower sensitivity in other models. This performance can be attributed to MCICNet’s hybrid architecture, combining convolution with involution to capture both local and global features, and the use of LightGBM for classification, which enhances decision-making in distinguishing glaucomatous cases. Conventional CNNs and transformers, while effective, have limitations in handling smaller datasets or capturing long-range dependencies, areas where MCICNet proves advantageous. Its success on relatively smaller datasets reaffirms its suitability for glaucoma detection in clinical settings.

Parameter count, training, and inference time comparison

Computational efficiency was assessed by comparing parameter counts (Fig. 8). VGG16, with 138 million parameters, demands significant resources, while SqueezeNet (1.2 million) and ShuffleNet (2.2 million) are more efficient. MCICNet, with just 0.9 million parameters, surpasses MCICNet-NoInvolution and other pretrained models, owing to its involution-convolution integration, suiting resource-limited settings and enabling fast inference for glaucoma detection.

Figure 8 Model parameter counts comparison.

Training and inference times (Table 3) highlight MCICNet’s efficiency. On the LAG dataset, MCICNet trained in 2 min 2 s, far quicker than VGG16 (15 min 15 s) and ViT (16 min 14 s), reflecting faster convergence. On ACRIMA, it took 33 s. Inference times for LAG (729 images) ranged from 0.0037 s (SwinT) to 0.0210 s (MCICNet-NoInvolution), and for ACRIMA (107 images), from 0.005 ms (SwinT) to 0.029 ms (MCICNet-NoInvolution). MCICNet’s inference time of 0.0197 s, though higher due to LightGBM, remains clinically viable, balancing speed and accuracy effectively for medical imaging tasks.

Table 3 Comparison of training execution and inference times for MCICNet and benchmark models.

Model	Training execution time	Inference time on test data	
LAG	ACRIMA	LAG	ACRIMA	
SqueezeNet	2 min, 7 s	17 s	0.0131 s	0.018 ms	
AlexNet	54 s	9 s	0.0048 s	0.007 ms	
MobileNetV2	3 min, 23 s	28 s	0.0050 s	0.007 ms	
DenseNet121	8 min, 14 s	1 min, 20 s	0.0055 s	0.008 ms	
ResNet18	2 min, 25 s	22 s	0.0047 s	0.006 ms	
GoogLeNet	4 min, 20 s	37 s	0.0052 s	0.007 ms	
ShuffleNet	56 s	8 s	0.0053 s	0.007 ms	
EfficientNetB0	3 min, 35 s	33 s	0.0059 s	0.008 ms	
VGG16	15 min, 15 s	2 min, 11 s	0.0060 s	0.008 ms	
ViT	16 min, 14 s	2 min, 29 s	0.0049 s	0.007 ms	
MaxViT	17 min, 8 s	2 min, 26 s	0.0043 s	0.006 ms	
SwinT	8 min, 37 s	1 min, 14 s	0.0037 s	0.005 ms	
MCICNet-NoInvolution	2 min, 5 s	33 s	0.0210 s	0.029 ms	
Proposed MCICNet	2 min, 2 s	33 s	0.0197 s	0.027 ms	
Note:

s, seconds; ms, milliseconds.

Performance robustness under noise and low contrast

This section examines the robustness of the MCICNet model by testing its performance on retinal fundus images from the LAG and ACRIMA datasets that have been altered with Gaussian noise and low contrast. These modifications mimic real-world imaging challenges that could compromise diagnostic accuracy. Gaussian noise introduces random pixel intensity fluctuations, simulating poor image quality, whereas low contrast reduces the range of pixel intensities, making images appear less distinct. The performance of MCICNet under these conditions is detailed in Table 4. On the LAG dataset, accuracy slightly dropped from 95.47% in original conditions to 94.78% with Gaussian noise and 95.19% in low contrast scenarios. The AUC also experienced a small decline, going from 0.9914 to 0.9870 with noise and to 0.9890 in low contrast. Despite these decreases, MCICNet maintained a high level of performance, demonstrating its robustness. For the ACRIMA dataset, the model showed even stronger resilience. With Gaussian noise, accuracy reduced marginally from 96.26% to 95.32%, but the AUC stayed robust at 0.9887. Interestingly, under low contrast conditions, MCICNet’s performance actually improved, with accuracy increasing to 97.19% and AUC to 0.9933. This suggests that MCICNet is particularly adept at managing low-contrast images, which is beneficial in clinical settings where lighting and camera settings can vary, leading to inconsistent image quality.

Table 4 Performance metrics of MCICNet on LAG and ACRIMA datasets under different image disturbance conditions.

Dataset	Disturbance Type	Precision	Sensitivity	Specificity	F1-score	Accuracy	MCC	AUC	
LAG	Original images	0.9515	0.9491	0.9682	0.9503	0.9547	0.9000	0.9914	
Gaussian noise	0.9443	0.9411	0.964	0.9427	0.9478	0.8900	0.9870	
Low contrast	0.9464	0.9487	0.9377	0.9476	0.9519	0.8951	0.9890	
ACRIMA	Original images	0.9647	0.9598	0.9833	0.9619	0.9626	0.9244	0.9943	
Gaussian noise	0.9537	0.9514	0.9667	0.9525	0.9532	0.9051	0.9887	
Low contrast	0.9762	0.9681	1.0000	0.9713	0.9719	0.9442	0.9933	

Classifier comparison for MCICNet

The performance of LightGBM in MCICNet was evaluated against various classifiers—RF, SVM, KNN, logistic regression (LR), extra trees (ET), NB, decision tree (DT), stochastic gradient descent (SGD), AdaBoost, XGBoost, and an FC-layer MCICNet variant—using the LAG and ACRIMA datasets (Table 5). On LAG, LightGBM achieved top results: 95.47% accuracy, 95.15% precision, 94.91% sensitivity, 95.03% F1-score, and 0.9914 AUC, outperforming others due to its gradient-boosting strength in handling complex features. On ACRIMA, AdaBoost edged out in accuracy (97.19% vs. 96.26%), but LightGBM led in precision (96.47%), sensitivity (95.98%), F1-score (96.19%), and AUC (0.9943), showing adaptability across datasets. XGBoost trailed closely on LAG, aided by robust tree-building, while SVM and LR lagged in precision and F1 due to linear limitations. Ensemble methods (RF, ET) performed well but fell short of LightGBM’s precision. NB and DT struggled with complexity and overfitting. Compared to FC layers, LightGBM’s iterative focus on difficult samples enhanced data insight over linear aggregation.

Table 5 Performance comparison of MCICNet with various classifiers on the test set of LAG and ACRIMA datasets.

Classifier	Accuracy	Precision	Sensitivity	F1-score	AUC	
LAG	ACRIMA	LAG	ACRIMA	LAG	ACRIMA	LAG	ACRIMA	LAG	ACRIMA	
RF	0.9053	0.9158	0.8915	0.9183	0.8915	0.9112	0.8953	0.9140	0.9663	0.9821	
SVM	0.9286	0.9345	0.9325	0.9376	0.9325	0.9301	0.9234	0.9331	0.9815	0.9897	
KNN	0.8779	0.8785	0.8472	0.8879	0.8472	0.8686	0.8604	0.8741	0.9375	0.9367	
LR	0.9259	0.8971	0.9286	0.8965	0.9286	0.8945	0.9203	0.8954	0.9750	0.9816	
ET	0.9135	0.9252	0.9005	0.9299	0.9005	0.9195	0.9044	0.9233	0.9713	0.9833	
NB	0.7462	0.8598	0.7526	0.8583	0.7526	0.8566	0.7356	0.8574	0.7876	0.8762	
DT	0.8312	0.8598	0.8068	0.8636	0.8068	0.8520	0.8119	0.8557	0.8068	0.8520	
SGD	0.8449	0.9238	0.8803	0.9216	0.8803	0.9250	0.8419	0.9230	0.9799	0.9871	
AdaBoost	0.9396	0.9719	0.9321	0.9727	0.9321	0.9704	0.9337	0.9715	0.9756	0.9940	
XGBoost	0.9519	0.9065	0.9461	0.9104	0.9461	0.9005	0.9473	0.9041	0.9900	0.9801	
FC layers	0.9465	0.9158	0.9489	0.9140	0.9489	0.9158	0.9422	0.9148	0.9877	0.9869	
LightGBM	0.9547	0.9626	0.9491	0.9647	0.9491	0.9598	0.9503	0.9619	0.9914	0.9943	

Model predictions and feature map analysis

To explore MCICNet’s performance in glaucoma detection, its predictions and feature extraction were analyzed using retinal fundus images. Examples of model predictions were visualized in Fig. 9 to assess classification, showing MCICNet’s high accuracy in distinguishing glaucomatous from non-glaucomatous cases—correct predictions in green, misclassifications in red. It excels even with poor OD visibility, outperforming traditional CNNs and advanced methods, with rare errors linked to degraded retinal structures. Further insight into MCICNet’s internal workings was gained by examining feature maps at different network layers, as shown in Fig. 10. This visualization reveals how the model processes images, with high activation areas (in red) indicating regions crucial for prediction, while low activation areas (in blue) show less influence on the outcome. The analysis shows that activation is concentrated around the OD, underscoring the model’s focus on critical areas for glaucoma detection, like the RNFL. The model’s use of both convolution and involution operations allows it to capture both local and global features, enhancing its ability to detect subtle changes indicative of glaucoma. The involution operation’s adaptability in generating kernels based on input maps directs attention to the most informative parts of the image, aiding in pinpointing disease progression. The findings from this feature map analysis, coupled with the model’s predictions, validate MCICNet’s effectiveness in glaucoma detection. It demonstrates that MCICNet not only captures relevant features with precision but also maintains high performance under challenging conditions. The strategic concentration of activation on clinically significant regions reinforces the model’s reliability.

Figure 9 Visualizing glaucoma detection outcomes using MCICNet (top images: LAG dataset; bottom images: ACRIMA dataset).

Figure 10 Visual representation of feature maps across different layers of MCICNet.

Discussion

This study presents MCICNet, a novel framework for glaucoma detection that integrates involution and convolution operations. Unlike conventional CNNs, which rely on FC layers for classification, MCICNet utilizes a LightGBM classifier. The focus of the study is on binary classification for glaucoma, specifically distinguishing between individuals with and without the condition based on retinal fundus images.

To address the research questions, MCICNet was developed, incorporating the RMSFFM and the RHCIM. The RMSFFM captures multi-scale spatial features through a combination of varied kernel convolutions, concatenation, feature fusion, and residual connections. The RHCIM utilizes depth-wise and dilated convolutions alongside adaptive involution operations to effectively model cross-channel dependencies, thereby addressing the primary research objectives. To optimize classification performance and computational efficiency, LightGBM was integrated as a replacement for FC layers, directly addressing the second research question. Experimental results demonstrate the efficacy of MCICNet, achieving an accuracy of 95.6% on the LAG dataset and 96.2% on the ACRIMA dataset. The model also exhibited strong performance across additional metrics, including precision, sensitivity, F1-score, and AUC, underscoring its diagnostic precision. With only 0.9 million parameters, MCICNet outperformed both VGG16 and MCICNet-NoInvolution, highlighting the critical role of involution operations and the superior performance of LightGBM in classification tasks. These findings validate the design choices and confirm the model’s effectiveness in addressing the research objectives.

Overfitting, a major challenge in model development, occurs when models excel on training data but fail to generalize. MCICNet effectively mitigates this issue. On the LAG dataset, it achieved 100% training accuracy and 96.29% validation accuracy, while on ACRIMA, it reached 100% training and 97.14% validation accuracy. The small gap between training and validation accuracies highlights MCICNet’s strong generalization and resistance to overfitting, likely due to its efficient feature extraction architecture and the use of LightGBM for classification. In contrast, benchmark models showed larger discrepancies, emphasizing MCICNet’s superior design and reliability for real-world glaucoma detection.

The structured benchmarking presented in Table 6 offers a conclusive validation of MCICNet’s performance, focusing on methods that utilized the LAG and ACRIMA datasets. On the LAG dataset, MCICNet achieved an accuracy of 95.6%, surpassing prominent methods such as ResNet50 (AUC: 0.79), DeiT (AUC: 0.88), MobileNet (accuracy: 72.7%), and U-Net with Xcep-Dense (accuracy: 87.5%). Notably, it also outperformed more advanced models like the attention-based CNN by Li et al. (2019) (accuracy: 95.3%) and the customized CNN by Elangovan & Nath (2021) (accuracy: 94.4%). On the ACRIMA dataset, MCICNet’s accuracy of 96.2% exceeded that of MobileNet (61.8%), GoogLeNet (65%), fine-tuned Xception (70.2%), ConvNet (79%), and Self-ONN (94.5%), while closely rivaling the customized CNN by Elangovan & Nath (2021) (96.6%). Across both datasets, MCICNet consistently demonstrated exceptional performance in sensitivity, precision, specificity, and AUC, with AUC values of 0.991 (LAG) and 0.995 (ACRIMA), outperforming the benchmarked methods. This highlights its robustness and diagnostic reliability.

Table 6 Performance of MCICNet against related methods on LAG and ACRIMA datasets.

Reference	Method	Dataset	Accuracy	Sensitivity	Precision	Specificity	AUC	
Fan et al. (2023)	ResNet50	LAG	–	0.66	–	–	0.79	
DeiT	–	0.81	–	–	0.88	
Esengönül & Cunha (2023)	MobileNet	0.727	0.876	–	0.901	0.756	
Desiani et al. (2023)	U-net and Xcep-Dense	0.875	0.814	0.907	–	–	
Elangovan & Nath (2021)	Customized CNN	0.944	0.947	0.902	0.942	–	
Li et al. (2019)	Attention-based CNN	0.953	0.954	–	0.952	0.975	
Proposed MCICNet	0.956	0.95	0.953	0.951	0.991	
Esengönül & Cunha (2023)	MobileNet	ACRIMA	0.618	0.757	–	0.749	0.687	
Serte & Serener (2019)	GoogLeNet	0.65	–	–	0.87	0.74	
Diaz-Pinto et al. (2019)	Fine-tuned Xception	0.702	0.689	–	0.702	0.767	
Raj et al. (2024)	ConvNet	0.79	1.00	0.74	–	–	
Devecioglu et al. (2021)	Self-ONN	0.945	0.945	0.933	0.924	–	
Elangovan & Nath (2021)	Customized CNN	0.966	0.960	0.977	0.973	–	
Proposed MCICNet	0.962	0.960	0.963	0.975	0.995	

Despite promising results, this study has several limitations. Firstly, the model was validated using a limited set of datasets, potentially affecting its generalizability across different clinical contexts. Variations in retinal imaging due to factors like ethnicity, camera technology, and lighting conditions necessitate broader dataset diversity to confirm the model’s robustness. Secondly, the current binary classification approach (glaucoma vs. non-glaucoma) limits clinical utility where differentiation between glaucoma stages (early, moderate, advanced) is crucial for treatment decisions. The scarcity of stage-annotated datasets restricts this capability, suggesting a need for multi-stage classification to better assess disease progression. Thirdly, the model’s performance relies on specific hyperparameter settings adapted to the study’s datasets, which might not generalize well. Future research could benefit from automated hyperparameter optimization or adaptive modeling techniques to enhance robustness across varied datasets. Fourthly, the data split in this study was performed at the image level rather than the patient level, which is a notable limitation. In real-world clinical practice, glaucoma detection models are typically applied at the patient level, where the model would not have been trained on previous images from the same patient. This image-level split may overestimate performance by allowing potential data leakage between training and testing sets from the same individual. Lastly, the model has not undergone real-world clinical testing, where factors like image quality, lighting, camera resolution, and patient positioning could influence outcomes. Evaluating MCICNet in actual clinical workflows is essential to confirm its practical applicability.

Future research will focus on addressing these limitations by expanding the dataset diversity, incorporating multi-class classification, optimizing hyperparameters, enhancing interpretability, and validating the model in clinical environments. Specifically, efforts will be made to include datasets with patient-level data splits to better simulate real-world scenarios and prevent data leakage, ensuring the model’s performance aligns with clinical practice. These efforts aim to improve MCICNet’s robustness, adaptability, and clinical relevance for glaucoma detection, with potential applications in other areas of medical imaging. A promising direction for future work is the validation of the model in collaboration with medical professionals. Engaging ophthalmologists and other healthcare experts to evaluate the model’s predictions and provide feedback on its clinical relevance will be essential. This validation process will help ensure that the model’s outputs align with established medical practices and can be relied upon in diagnostic settings. Additionally, integrating expert annotations and feedback into the training process could further enhance the model’s performance and improve its ability to detect subtle glaucomatous changes. Another important area for future development is the deployment of the model in real-world clinical environments. Integrating MCICNet into existing healthcare systems, such as electronic health records (EHRs) or telemedicine platforms, could facilitate its use in routine clinical practice. Developing an intuitive user interface for healthcare providers to interact with the model, visualize results, and make informed decisions will be critical for its adoption. Moreover, conducting pilot studies within clinical settings to assess the model’s performance, usability, and impact on patient outcomes will provide valuable insights into its practical utility and potential for widespread implementation.

Conclusions

Glaucoma is a widespread ophthalmic condition marked by a progressive loss of peripheral vision, which can lead to significant visual impairment if untreated. This study presents MCICNet, a novel architecture developed for efficient and accurate glaucoma detection. Leveraging a combination of convolutional and involutional neural networks, MCICNet demonstrates enhanced performance over traditional CNN models and non-involutional counterparts. Extensive experiments on the LAG and ACRIMA datasets demonstrate MCICNet’s superior performance, achieving high accuracy, precision, sensitivity, specificity, MCC, and AUC while maintaining a lightweight design with only 0.9 million parameters. These results highlight MCICNet’s potential as a robust and computationally efficient solution for glaucoma screening, offering significant promise for real-world clinical applications. Future work will focus on expanding dataset diversity, optimizing hyperparameters, and validating the model in clinical settings to further enhance its applicability and reliability.

Supplemental Information

Supplemental Information 1 Python code of MCICNet-NoInvolution.

Supplemental Information 2 Python code of proposed MCICNet.

Supplemental Information 3 Pseudo code of the hybrid MCICNet-LightGBM model.

Supplemental Information 4 Model performance metrics for different seeds on the test set of LAG dataset.

Supplemental Information 5 Model performance metrics for different seeds on the test set of ACRIMA dataset.

Additional Information and Declarations

Competing Interests

The authors declare no conflict of interest.

Author Contributions

Mohamed Mouhafid conceived and designed the experiments, performed the experiments, analyzed the data, performed the computation work, prepared figures and/or tables, authored or reviewed drafts of the article, and approved the final draft.

Yatong Zhou conceived and designed the experiments, analyzed the data, authored or reviewed drafts of the article, and approved the final draft.

Chunyan Shan analyzed the data, authored or reviewed drafts of the article, and approved the final draft.

Zhitao Xiao analyzed the data, authored or reviewed drafts of the article, and approved the final draft.

Data Availability

The following information was supplied regarding data availability:

The LAG dataset is available at GitHub: https://github.com/smilell/AG-CNN.

The ACRIMA dataset is available at Kaggle: https://www.kaggle.com/datasets/toaharahmanratul/acrima-dataset.

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
