# Peer review of "Towards efficient glaucoma screening with modular convolution-involution cascade architecture"

_PeerJ Computer Science, doi:10.7717/peerj-cs.2844_

## Round 0.1 · original submission · Major Revisions

Dear authors,

You are advised to critically respond to all comments point by point when preparing an updated version of the manuscript and while preparing for the rebuttal letter. Please address all comments/suggestions provided by reviewers, considering that these should be added to the new version of the manuscript.

Kind regards,
PCoelho

·

Basic reporting

Author developed CNN architecture, MCICNet, to enhance cross-channel feature extraction for automated glaucoma detection from retinal fundus images. The following things neet to be addressed for better understanding.

Clear difference between the available literature and previous works are missing, Authors are asked to provide the limitations of the previous correlated works and then link those limitations to the current ideas and contributions of the current work.

No preprocessing step is used prior to classification.

MCC can be computed, Which provides a better decision for unbalanced data.

The interpretability of the model is lagging. Advantages and shortcoming of the network are not written.
Can the code and data be shared for reproducing the results?

How the model will behave in presence of noise and low contrast image.

Experimental design

NA

Validity of the findings

Need to be validated with some other database.

Reviewer 2 ·

Basic reporting

The basic reporting is fairly comprehensive. However, additional detail on the clinical definition of glaucoma used in the study might be provided, including how the appearance of the retina in fundus images is used to determine presence of glaucoma.

Experimental design

Please see the additional comments section.

Validity of the findings

The findings appear mostly valid, with some concerns covered in the additional comments section.

Additional comments

1. In Section 2.3, it is stated that LightGBM is used as an advanced classifier to analyze extracted feature vectors. However, CNN models are typically trained end-to-end, i.e. the output loss for backpropagation is computed based on the target prediction (i.e. the target glaucoma class). However, Figure 1 shows the LGBM module inserted right after the RHCIM4 layer.

It might thus be clarified as to what objective was used for MCICNet, for feature extraction, especially since Table 3 suggests that various other classifiers were tried for the final classification.

2. In Section 2.5, it might be clarified as to whether the 70:15:15 split for training, validation and test sets is standard, or randomly determined by the authors. Also, it might be clarified as to whether this split is at the patient, eye or image level.

3. From Table 2, the best-performing model other than the proposed MCICNet is VGG16, which is the oldest of all comparison models other than AlexNet. These results might be discussed in further detail, given that more-recent models (i.e. ResNet, DenseNet, SqueezeNet, EfficientNet etc.) are generally expected to outperform older models. In particular, any pretraining/settings/hyperparameter optimization for all these CNN models might be described.

4. Specificity and AUC metrics should be added to Table 2. If required, the table can be split into two subtables covering validation and test results respectively. Likewise, specificity metrics should be added to Table 3.

5. Example input images for various glaucoma classes, for both true and false predictions, should be included. If possible, saliency heatmaps should also be included as figures.

Reviewer 3 ·

Basic reporting

Good.
Flow / structure can be improved.

Experimental design

Good

Validity of the findings

Can elaborate

Additional comments

With the advancement of deep learning, several studies have addressed classification of eye diseases using fundus images, the novelty of this study falls under the area of utilizing a combination of convolution and involution operations in a cascade structure that allows to capture inter-channel dependencies within the feature extraction process.
The paper is well-written and structured. The methodology is described well.

The following improvements are suggested.

1. It would be better to highlight the main research questions in the introduction section, and describe in the Discussion section, how you fulfilled those in the study, by referring to the applied methods and the obtained results.

2. Use of / citation of references, does not show the competency in the critical analysis of the literature. That is, you have continuously cited references, and it seems you have not done a thorough analysis. For example, a give reference can be cited in many places and for a given fact, many references can be cited.


3. Better to include a new Section 2: related work, describe the techniques used to segment and classify retinal fundus images, with more latest studies with different categories such that segmentation only, classification only, segmentation followed by classification, for different multi-tasks you have considered.


4. Scientific contribution is good. It would be better to include a pseudocode for the approach, that support reproducibility.


5. Describe the implementation details such as used libraries and the hardware specification (CPU/ GPU)

6. Include more evaluation metrics such as the learning curves (training and validation/ accuracy and loss)

7. Discuss the issues and limitations (such as overfitting/ underfitting) in the training model and how you can address those.

8. For all the results figures and tables, clearly explain the observations and the decisions that can be taken by those observations.

9. Include a comparison table in the discussion, stating the new features and results of the proposed model, in comparison to the latest such studies.

10. Discuss the future work and possibility of deploying the proposed model as a real world application in clinical settings. Because most of the AI based models are not implemented for actual practice.

11. What is the possibility of validating this approach with medical experts.

12. Since this study is going to be published in 2024, include more studies from the last 3 years, 2022, 2023 and 2024.

·

Basic reporting

No Comment.

Experimental design

Line 363: We appreciate the detailed description of each benchmark model. The authors should explicitly declare the input dimensions and training setup for each benchmark model.

Line 335: In this section, the authors have introduced LightGBM in detail. However, it should be made clear if and how LightGBM is jointly optimized with the CNN layers before it via gradient descent, which is the cornerstone of deep learning. If LightGBM is trained/fitted separately from the MCICNet, what y label is used to train MCICNet for its feature extraction capability?

Line 496: The authors should declare the seed used during training. To demonstrate robustness against randomness, the authors can consider report the average performances across more than one seed, compared with other benchmark models.

Table 3: How many fully connected layers are used in the variation of MCICNet? If only one layer is used, its linear capacity will likely not able to compete with other classifiers such as LightGBM.

Validity of the findings

Line 666: To further demonstrate the efficiency and of proposed network and its suitability for deployment in resource-constrained environments (in addition to pure parameter count), benchmark GPU and/or CPU runtimes at inference stage for the different models and compare.

Line 666: We appreciate the level of detail described for the different modules of the model, including LightGBM. It would be great if the authors could show the generalizability of such architecture on one or more additional (open) datasets (such as MIMIC), compared with other benchmark models.

Table 6: It is not made clear how this table contributes to the main aims of the paper. The models in the table are trained on different datasets using different setups in different studies, and it is not clear how they are comparable with MCICNet.

Additional comments

Line 407: Since the publication of “An Image is Worth 16x16 Words: Transformers for Image Recognition at Scale” in 2020, transformer-based vision models have demonstrated dominant performances across vision tasks at scale and is widely adapted to different domains. The authors should consider including a variant of transformer-based vision models as benchmark comparison as well. A ViT-S model takes 86 million parameters (less than VGG 16), and some versions have Scaled Dot Product Attention (SDPA) implementation, which is both memory and time efficient. It will be more convincing if the proposed architecture can should performance and efficiency superiority over ViT models, as it is the new generation of base model instead of CNNs.

---

## Round 0.2 · Minor Revisions

Dear authors,

Thanks a lot for your efforts to improve the manuscript.

Nevertheless, some concerns are still remaining that need to be addressed.

Like before, you are advised to critically respond to the remaining comments point by point when preparing a new version of the manuscript and while preparing for the rebuttal letter.

Kind regards,
PCoelho

Reviewer 2 ·

Basic reporting

The basic reporting is acceptable.

Experimental design

As it is now clarified that the data split is at the image level, this should be acknowledged as additional limitation since in real-life practice, application for glaucoma and other medical imaging models is more commonly at the patient level (i.e. the model would not have been trained on previous images from the test patient).

Validity of the findings

The findings appear valid.

Additional comments

N/A

Reviewer 3 ·

Basic reporting

Good

Experimental design

Good

Validity of the findings

Improved

·

Basic reporting

The authors successfully addressed reviewer concerns.

Experimental design

The authors successfully addressed reviewer concerns.

Validity of the findings

The authors successfully addressed reviewer concerns.

Additional comments

The authors successfully addressed reviewer concerns.

---

## Round 0.3 · accepted · Accept

Dear authors, we are pleased to verify that you meet the reviewer's valuable feedback to improve your research.

Thank you for considering PeerJ Computer Science and submitting your work.

Kind regards
PCoelho

Reviewer 2 ·

Basic reporting

The basic reporting is acceptable.

Experimental design

Concerns about experimental design have been resolved.

Validity of the findings

The findings appear valid.

Additional comments

N/A